# Endothelial Cell Markers Are Inferior to Vascular Smooth Muscle Cells Markers in Staining Vasa Vasorum and Are Non-Specific for Distinct Endothelial Cell Lineages in Clinical Samples

**DOI:** 10.3390/ijms24031959

**Published:** 2023-01-19

**Authors:** Victoria Markova, Leo Bogdanov, Elena Velikanova, Anastasia Kanonykina, Alexey Frolov, Daria Shishkova, Anastasia Lazebnaya, Anton Kutikhin

**Affiliations:** Department of Experimental Medicine, Research Institute for Complex Issues of Cardiovascular Diseases, 6 Sosnovy Boulevard, Kemerovo 650002, Russia

**Keywords:** endothelial cells, vascular smooth muscle cells, vasa vasorum, arterioles, venules, capillaries, microcirculation, α-SMA, SM-MHC, endothelial heterogeneity

## Abstract

Current techniques for the detection of vasa vasorum (VV) in vascular pathology include staining for endothelial cell (EC) markers such as CD31 or VE-cadherin. However, this approach does not permit an objective assessment of vascular geometry upon vasospasm and the clinical relevance of endothelial specification markers found in developmental biology studies remains unclear. Here, we performed a combined immunostaining of rat abdominal aorta (rAA) and human saphenous vein (hSV) for various EC or vascular smooth muscle cell (VSMC) markers and found that the latter (e.g., alpha smooth muscle actin (α-SMA) or smooth muscle myosin heavy chain (SM-MHC)) ensure a several-fold higher signal-to-noise ratio irrespective of the primary antibody origin, fluorophore, or VV type (arterioles, venules, or capillaries). Further, α-SMA or SM-MHC staining allowed unbiased evaluation of the VV area under vasospasm. Screening of the molecular markers of endothelial heterogeneity (mechanosensitive transcription factors KLF2 and KLF4, arterial transcription factors HES1, HEY1, and ERG, venous transcription factor NR2F2, and venous/lymphatic markers PROX1, LYVE1, VEGFR3, and NRP2) have not revealed specific markers of any lineage in hSV (although KLF2 and PROX1 were restricted to venous endothelium in rAA), suggesting the need in high-throughput searches for the clinically relevant signatures of arterial, venous, lymphatic, or capillary differentiation.

## 1. Introduction

The blood vessel wall is supplied with oxygen and nutrients through the network of vasa vasorum (VV) which are indispensable for vascular physiology but are also expanded at hypoxic conditions or inflammation [1,2]. As other vessels, VV can have arterial, venous, and capillary identity and form multiple branches as a result of sprouting angiogenesis, eventually deploying an intricate meshwork across the entire wall of nourished blood vessels [3,4,5,6,7]. Growth of VV is stimulated by hypoxia and acidification and is mediated by hypoxia-inducible factor 1-alpha (HIF-1α), vascular endothelial growth factor (VEGF), and basic fibroblast growth factor (bFGF) expressed by vascular smooth muscle cells (VSMCs) located within the tunica media as well as by adventitial fibroblasts and macrophages [8,9,10]. Anatomically, VV represent a natural route for the migration of circulating immune cells into the tunica adventitia, perivascular adipose tissue, and neointima, and increased numbers and size of VV correlate with neointimal formation in restenosis and atherosclerosis scenarios [1,2,3,4,5,6,7]. Tissue-engineered vascular grafts (TEVGs) also contain significant amounts of VV because of intensive biosynthetic processes demanding high income of energy and nutrients, and development of the VV network in TEVGs ensures migration of monocytes/macrophages responsible for the polymer biodegradation and mesenchymal cells (including mesenchymal stem cells and VSMC progenitors) producing the extracellular matrix [11,12]. The contribution of distinct VV specifications into inflammation or regeneration remains unclear to date, although several markers of arterial and venous differentiation have been proposed [13,14,15].

Currently, identification and evaluation of the VV is carried out by an immunostaining to cluster of differentiation (CD)31 (also termed platelet and endothelial cell (EC) adhesion molecule 1, PECAM1), vascular endothelial (VE) cadherin, and von Willebrand factor (vWF) [16,17]. In contrast, vascular endothelial growth factor receptor 3 (VEGFR3), lymphatic vessel endothelial hyaluronan receptor 1 (LYVE1), and Prospero homeobox protein 1 (PROX1) are believed to be the specific lymphatic vessel markers [18,19]. However, immunostaining to EC markers has a number of inherent drawbacks such as inability to assess vascular geometry in the case of the shrinkage of the vessel lumen (a frequent consequence of vasospasm) or insufficient signal intensity because of relatively small size of the ECs. In addition, specific markers of EC lineages, which have been found and successfully verified in the developmental biology models, have not been properly tested in clinical samples such as blood vessels excised during the coronary artery bypass graft surgery.

Here, we showed that VSMC markers (i.e., contractile proteins) demonstrate excellent signal intensity in addition to the perfect sensitivity and specificity while staining VV within the human saphenous vein (SV). Such staining (e.g., by α smooth muscle actin (α-SMA) or smooth muscle myosin heavy chain (SM-MHC) antibodies) ensures objective assessment of the vascular geometry and provides a significantly higher signal-to-noise ratio as compared to the conventional immunostaining against EC markers. Albeit, staining for CD31, VE-cadherin, and E26 transformation-specific (ETS)-related gene (ERG) transcription factor were highly specific for all EC lineages, and PROX1 and KLF2 transcription factors were exclusively expressed in venous ECs lining VV of the rat abdominal aorta (rAA), none of the screened markers of endothelial specification were restricted to any endothelial lineage in VV of the human SV (hSV). We suggest that VSMC markers are superior to EC markers when staining the microcirculation (e.g., VV), yet molecular discrimination of distinct VV types is challenging and requires a high-throughput proteomics analysis to find clinically relevant markers of endothelial lineages in the adult state.

## 2. Results

### 2.1. VSMC Markers Are Superior to EC Markers in VV Staining Applications

We first performed a routine hematoxylin and eosin (H&E) and Russell–Movat’s pentachrome staining to differentiate arterioles, venules, and capillaries within the hSV to make a gauge for the further molecular characterization of these blood vessels. Arterioles were distinguished by a dense internal elastic lamina (Figure 1A,B), while venules (Figure 1C,D) and capillaries (Figure 1E,F) were discriminated by the count of VSMC/pericyte layers around the EC monolayer. Although the internal elastic lamina has been vaguely visible at H&E staining (Figure 1A,C,E), it was clearly observable at Russell–Movat’s pentachrome staining (Figure 1B,D,F). In addition, the internal elastic lamina could be detected at immunofluorescence microscopy even without a specific antibody due to a significant autofluorescence in the blue channel, hence limiting the number of antibodies for the discrimination of blood vessel lineage to two (i.e., those emitting signals in red and green channels) instead of three.

Specific staining of the hSV for EC (CD31 and VE-cadherin) and VSMC (α-SMA and SM-MHC) markers revealed multiple VV in the tunica adventitia, consisting of a continuous EC monolayer and several concentric VSMCs (Figure 2A–L). Staining for CD31/PECAM1 yielded a stronger but less detailed signal than the staining for VE-cadherin, regardless of primary antibody origin (rabbit or mouse) or type of the fluorophore (Alexa Fluor 555 or 488) (Figure 2A–L). Fluorescent-labeled antibodies to CD31 demonstrated higher signal intensity whilst those to VE-cadherin permitted intercellular junctions to be delineated (Figure 2A–L). Notably, antibodies to CD31 stained both the lateral and basal surfaces of the ECs whereas antibodies to VE-cadherin highlighted exclusively cell–cell contacts at the lateral surfaces (Figure 2A–L). Staining of VSMCs by the antibodies to contractile proteins showed high intensity regardless of the selected marker (SM-MHC or α-SMA), primary antibody origin (rabbit or mouse), or type of the fluorophore (Alexa Fluor 555 or 488) (Figure 2A–L). Although SM-MHC expression is believed to be restricted to VSMCs in comparison with α-SMA which is also contained in the extracellular matrix, both of these markers were detected in VSMCs and the medial extracellular matrix (Figure 2A–L). Albeit, the arterioles (Figure 2A–D), venules (Figure 2E–H), and capillaries (Figure 2I–L) could be stained by both anti-EC and anti-VSMC antibodies, the latter showed a several-fold higher signal-to-noise ratio irrespective of the selected marker (SM-MHC or α-SMA), primary antibody origin (rabbit or mouse), or type of the fluorophore (Alexa Fluor 555 or 488) because of the larger VSMC size and higher number of layers (≥1, Figure 2A–L).

In contrast to anti-CD31 or anti-VE-cadherin staining, anti-SM-MHC or anti-α-SMA staining permitted a robust visualization of vascular geometry even in the case of vasospasm frequently affecting the capillaries during the excision of the blood vessel (Figure 3A–H). Therefore, we suggested staining against the contractile proteins (VSMC markers, e.g., SM-MHC or α-SMA) as an alternative and efficient option for the detection of the microcirculation as it outperforms the conventional approach (i.e., staining against the EC markers such as CD31 or VE-cadherin) regardless of the experimental conditions and selected antibody.

### 2.2. EC Markers Are Not Restricted to Any of the Lineages in Clinical Samples

We next investigated whether arterioles, venules, and capillaries bear specific markers of the respective EC lineage in the adult state. Having screened a number of markers reported from the developmental biology studies in rAA, we found Krüppel-like factor 2 (KLF2), a mechanosensitive transcription factor, and PROX1, a transcription factor of venous and lymphatic specification, as highly specific for venous ECs in the adventitial VV (Figure 4A–F). Further, hairy/enhancer-of-split related with YRPW motif protein 1 (HEY1), an arterial differentiation transcription factor, was detected exclusively in the capillaries (Figure 4G–I). ERG transcription factor was detected in all ECs (Figure 4J–L), KLF4 mechanosensitive transcription factor was found in VSMCs and pericytes (Figure 4M–O), and hairy and enhancer of split-1 (HES1) transcription factor was expressed in all vascular cell populations of the rAA (Figure 4P–R).

However, staining of hSV found KLF2 transcription factor restricted neither to venules nor to ECs (Figure 5A,B). Further, expression of HEY1 in the capillaries was moderate at best (Figure 5C), and PROX1 was not detected in any of the VV (Figure 5D–F). Other markers of venous and lymphatic differentiation such as LYVE1 (Figure 5G–I), VEGFR3 (Figure 5J–L), and neuropilin 2 (NRP2, Figure 5M–O) were not specific for ECs and were detected in all blood vessel types, though LYVE1 expression was higher than those of VEGFR3 and NRP2 (Figure 5G–I). Similar to the venous and lymphatic markers, KLF4 was also expressed in all vascular lineages (Figure 5P–R), whereas HES1 was not expressed in the VV within the hSV (Figure 5S–U). Similar to rAA, ERG transcription factor was specific for the ECs regardless of their lineage (Figure 5V–X) and therefore can be considered as a pan-endothelial marker in addition to CD31 and VE-cadherin. Albeit, nuclear receptor subfamily 2 group F member 2 (NR2F2, also termed chicken ovalbumin upstream promoter transcription factor 2, COUP-TFII) was exclusively (but still variably) expressed in venous VV (Figure 5S–X); immunohistochemical staining of the hSV for NR2F2 documented its expression across the whole vascular wall at all antibody dilutions (1:100, 1:200, and 1:500, Figure 6A–C). Even at 1:500 dilution, all VV within the hSV were positive for NR2F2 expression (Figure 6D).

Hence, canonical markers of arterial, venous, and lymphatic identity lacked their specificity in the clinical samples obtained from adult patients. Some of them (KLF2, KLF4, LYVE1, VEGFR3, NRP2, NR2F2, and ERG) were represented in all VV within the hSV (KLF2 and NR2F2, entire vascular wall; ERG, ECs; KLF4, LYVE1, VEGFR3, and NRP2, VSMCs), while others (PROX1, HEY1, and HES1) have not been detected or demonstrated a negligible expression (although the respective antibodies were successfully validated in rAA). Transcription factors of the arterial lineage (HES1 and ERG) were not specific for any of endothelial lineages in rAA and hSV, as HES1 was presented in all vascular cells (rAA) or was not expressed at all (hSV), and ERG served as a pan-endothelial marker in both of these vessels.

## 3. Discussion

In pathology, detection of the blood vessels including microcirculation (e.g., VV), is commonly conducted using immunohistochemical or immunofluorescence staining for a number of specific EC markers including CD31, VE-cadherin, VEGFR2, vWF, and CD34 [17,18,19,20], yet none of them is free from criticism. For instance, CD34 represents a marker of endothelial progenitor cells [21] and other stem cells [22]. Although vWF is abundant in the EC cytosol where it is stored in Weibel–Palade bodies, it is also located in the subendothelial extracellular matrix that is notable during staining of tissue-engineered vascular grafts [21,23]. In addition, vWF is found in clots because of its pro-thrombotic function [24,25]. VE-cadherin mediates intercellular adhesion and despite being specific for ECs, stains only cell–cell junctions and does not stain single ECs and apical or basal EC surfaces. CD31 (PECAM1) is expressed both in the ECs and platelets and hence can be detected in thrombi similar to vWF [26]. Another highly specific marker of the ECs is VEGFR2, but its level is in an order of magnitude lower than those of CD31 or VE-cadherin that substantially limits its sensitivity, in particular during the staining of bioartificial tissues (e.g., tissue-engineered vascular grafts) and xenogeneic implants (e.g., bioprosthetic heart valves or vascular patches for carotid endarterectomy). The most frequently applied EC markers are CD31 (which can differentiate ECs from platelets if it combines with any nuclear stain such as 4′,6-diamidino-2-phenylindole) and VE-cadherin, as both of them ensure intense and specific staining even at high antibody dilutions.

A major drawback of using anti-EC antibodies as microvessel markers is that ECs are prone to vasospasm that causes shrinkage of the vessel lumen and does not allow the proper assessment of its area, a key marker of inflammation. In this case, the microvessel can still be detected as a cluster of cells but unclear geometry is incompatible with the machine learning and automated detection algorithms which have recently become widespread in digital pathology. Hence, here we aimed to assess VSMC contractile proteins as alternative markers of the blood vessels. In comparison with the EC monolayer, VSMCs formed ≥ 1 concentric layer and occupied a significantly higher area providing a stronger signal from horseradish peroxidase or fluorescent-labeled antibodies. Since VSMCs comprise ≥ 80% of the microvessels, specific immunostaining permitted an adequate analysis of the vascular geometry even at vasospasm and ensured high-quality machine learning for the development of the artificial intelligence tools. Among the VSMC markers are SM-MHC, α-SMA, calponin, SM22α, and smoothelin, although SM-MHC and α-SMA are the most frequently used [27,28,29].

In our study, SM-MHC and α-SMA had the similar staining pattern and were exclusively expressed in VSMCs. Combined immunostaining for EC and VSMC markers confirmed excellent signal-to-noise ratio upon anti-SM-MHC or anti-α-SMA staining that furnished an opportunity for VV immunodetection and unbiased evaluation of blood vessel area irrespective of the section, antibody origin, fluorophore, or VV type (arterioles, venules, or capillaries). In addition to VSMCs, pericytes were also positively stained with anti-SM-MHC or anti-α-SMA antibodies, testifying to the versatility of these markers in terms of microvessel staining. This corresponds to the papers reporting an α-SMA expression by pericytes in the retina [30], brain [31,32], and lungs [33,34]. Albeit the repertoire of molecular expression varies in pericytes of different organs and tissues containing specialized capillaries, the pericyte expression of α-SMA suggests the applicability of VSMC contractile proteins to assess the microcirculation in the tissues and organs other than blood vessels. Importantly, neither α-SMA nor SM-MHC are detectable in ECs by immunofluorescence staining or quantitative polymerase chain reaction. 

Next, we attempted to find specific markers of arterioles, venules, and capillaries which could allow their discrimination upon immunostaining to develop the artificial intelligence tools for digital pathology. Canonical markers of the arterial lineage include Notch pathway transcription factors (HES1, HEY1, HEY2, and ERG) [35,36], whereas those specific for venous specification are NR2F2 transcription factor (COUP-TFII) [37] and neuropilin 2 (NRP2), a co-receptor to VEGF-A and VEGF-C [38]. Proteins restricted to the lymphatic identity are PROX1 transcription factor, LYVE1, an integral membrane glycoprotein recognizing hyaluronic acid, VEGFR3, a receptor for VEGF-C and VEGF-D, and a transmembrane glycoprotein podoplanin [39,40]. Markers of capillary differentiation remain obscured, largely because of variable anatomy and molecular profile of the capillaries in distinct organs.

Albeit, we tested a wide spectrum of molecular markers (mechanosensitive transcription factors KLF2 and KLF4, arterial transcription factors HES1, HEY1, and ERG, venous transcription factor NR2F2, and venous/lymphatic markers PROX1, LYVE1, VEGFR3, and NRP2) and some of them showed conclusive results in rats (i.e., KLF2 and PROX1 were exclusively found in rAA venules); however, they failed to demonstrate specificity for any of microvessel lineages in hSV. PROX1, HEY1, and HES1 have not been expressed in hSV whilst KLF2, KLF4, NR2F2, LYVE1, VEGFR3, and NRP2 were expressed in all VV. The expression of KLF4, LYVE1, VEGFR3, and NRP2 was restricted to VSMCs, although LYVE1 and VEGFR3 have been reported to be specific for lymphatic ECs [39,40] and NRP2 was ostensibly documented as a marker of venous ECs [19]. Notably, ERG transcription factor was specific for all EC specifications in rAA and hSV, suggesting this molecule as a reliable marker of endothelial differentiation in addition to CD31 and VE-cadherin.

For this study, we have used the frozen sections as they are typically used for the immunofluorescence staining whilst the paraffin-embedded sections are commonly employed for immunohistochemical analysis [41,42,43,44,45]. The disadvantages of utilizing paraffin-embedded sections for the immunofluorescence staining are: (1) masking of epitopes and high endogenous fluorescence caused by prolonged (24 h) formalin fixation, the drawbacks which are frequently irreversible [41,42,43,44,45]; (2) uncontrolled alteration of section orientation on slides which is incompatible with consecutive visualization, where all sections should be as similar to each other as possible; (3) relatively high number of section folds which can also lead to the loss of regions of interest. Further, immunofluorescence staining often requires less antibody dilutions as compared to immunohistochemical staining [41], is less sensitive to the background during the microscopy [41,42,43,44,45], is less cumbersome, and the visualization of two antigens simultaneously is significantly cheaper by the immunofluorescence approach. Due to these reasons, formalin-fixed paraffin-embedded tissues have not been employed for this study.

Hence, we suggest that Russell–Movat’s pentachrome staining, which highlights the elastic fibers unique for arterioles and allows to discern venules and capillaries by their geometry and wall composition, represents an optimal approach for the discrimination of the arterioles from venules and capillaries in machine learning applications. Specific molecular markers of blood vessel lineages in clinical samples remain unclear hitherto.

## 4. Materials and Methods

The segments of hSV (n = 3) were obtained during the coronary artery bypass graft surgery from the patients admitted to the Research Institute for Complex Issues of Cardiovascular Diseases (Kemerovo, Russia). The collection of clinical specimens was approved by the Local Ethical Committee of the Research Institute for Complex Issues of Cardiovascular Diseases (ethical approval code 37/2022, approved on 17 March 2022), and a written informed consent was provided by all study participants after receiving a full explanation of the study. The investigation was carried out in accordance with the Good Clinical Practice and a latest revision of Declaration of Helsinki (2013). Segments of rAA were excised from male Wistar rats weighing 250–300 g and 14–18 weeks of age. Animals (n = 3) were provided by the Research Institute for Complex Issues of Cardiovascular Diseases Core Facility. All procedures were carried out conforming to the European Convention for the Protection of Vertebrate Animals used for Experimental and Other Scientific Purposes (Strasbourg, France, 1986) and Directive 2010/63/EU of the European Parliament on the protection of animals used for scientific purposes, and were approved by the Local Ethical Committee of the Research Institute for Complex Issues of Cardiovascular Diseases (ethical approval code 38/2022, approved on 17 March 2022).

Upon the excision, blood vessels were briefly flushed with a physiological saline (Hematek, Tver, Russia), snap-frozen in the optimal cutting temperature medium (Tissue-Tek, 4583, Sakura Finetek, Tokyo, Japan), and cut on a cryostat (7 µm sections, Microm HM 525, Thermo Scientific, Waltham, MA, USA). For the H&E staining, tissues were fixed in 4% paraformaldehyde (158127, Sigma-Aldrich, Saint Louis, MO, USA) for 10 min, washed in a double distilled water for 15 min, and stained with hematoxylin (05-003, ErgoProduction, Saint Petersburg, Russia) and eosin (05-011, ErgoProduction, Saint Petersburg, Russia) as described in [34]. Russell–Movat’s pentachrome staining was performed using the commercially available kit (ab245884, Abcam) according to the manufacturer’s protocol. Coverslips were mounted with Vitrogel (HM-VI-A500, ErgoProduction, Saint Petersburg, Russia). Visualization was carried out by light microscopy (AxioImager.A1 microscope and EC Plan-Neofluar 20×/0.50 or EC Plan-Neofluar 40×/0.75 M27 objectives, Carl Zeiss, Oberkochen, Germany).

For the immunofluorescence staining, the sections were fixed in 4% paraformaldehyde (158127, Sigma-Aldrich, Saint Louis, MO, USA) for 10 min, permeabilized in Triton X-100 (T8787, Sigma-Aldrich, Saint Louis, MO, USA) for 15 min, and blocked in 1% bovine serum albumin (P091E, PanEco, Moscow, Russia) for 1 h to prevent non-specific binding. Sections were then incubated for 16 h at 4 °C with the following antibodies:

rAA:(1)KLF2 (1:200, NBP2-61812, Novus Biologicals, Centennial, CO, USA) as a single stain and in combination with PROX1 (1:50, ab199359, Abcam, Cambridge, UK), HEY1 (1:50, ab154077, Abcam, Cambridge, UK), ERG (1:50, ab92513, Abcam, Cambridge, UK), or KLF4 (1:50, ab215036, Abcam, Cambridge, UK);(2)HES1 (1:50, ab108937, Abcam, Cambridge, UK) as a single stain.

hSV:(1)CD31 (1:100, ab28364 or 1:500, ab9498, Abcam, Cambridge, UK) in combination with SM-MHC (1:250, ab683 or 1:250, ab224804, Abcam, Cambridge, UK);(2)VE-cadherin (1:250, 2500S, Cell Signaling Technology, Danvers, MA, USA, or 1:750, ab33168, Abcam, Cambridge, UK) in combination with α-SMA (1:250, ab7817, Abcam, Cambridge, UK);(3)KLF2 (1:200, NBP2-61812, Novus Biologicals, Centennial, CO, USA) as a single stain and in combination with HEY1 (1:100, ab154077, Abcam, Cambridge, UK);(4)PROX1 (1:100, ab199359, Abcam, Cambridge, UK) as a single stain;(5)CD31 (1:500, ab9498, Abcam, Cambridge, UK) in combination with LYVE1 (1:100, ab14917, Abcam, Cambridge, UK), VEGFR3 (1:100, ab27278, Abcam, Cambridge, UK), or NRP2 (1:100, ab185710, Abcam, Cambridge, UK);(6)KLF4 (1:100, ab215036, Abcam, Cambridge, UK) as a single stain;(7)NR2F2 (1:50, ab41859, Abcam, Cambridge, UK) in combination with HES1 (1:50, ab108937, Abcam, Cambridge, UK) or ERG (1:50, ab92513, Abcam, Cambridge, UK).

The next day, sections were further treated with donkey anti-rabbit or anti-mouse pre-adsorbed Alexa-Fluor-488-conjugated (1:500, ab150061 or ab150109, Abcam, Cambridge, UK) and donkey anti-rabbit or anti-mouse pre-adsorbed Alexa-Fluor-555-conjugated (1:500, ab150062 or ab150110, Abcam, Cambridge, UK) secondary antibodies for 1 h at room temperature. Nuclear counterstaining was performed with DAPI (10 μg/mL, D9542, Sigma-Aldrich, Saint Louis, MO, USA) for 30 min at room temperature. At all stages, washing was conducted with 0.1% phosphate-buffered saline (60201, Pushchino Laboratories, Pushchino, Russia) solution of Tween-20 (P9416, Sigma-Aldrich, Saint Louis, MO, USA). Coverslips were mounted with ProLong Gold Antifade (P36934, Thermo Fisher Scientific, Waltham, MA, USA). Slides were examined by confocal laser scanning microscopy (LSM 700, Carl Zeiss, Oberkochen, Germany).

Immunohistochemical staining was performed using Novolink Polymer Detection Systems Novocastra (RE7150-CE, Leica Biosystems, Wetzlar, Germany) according to the manufacturer’s protocol. Sections were incubated with a mouse antibody to NR2F2 (1:500, ab41859, Abcam, Cambridge, UK) for 16 h at 4 °C. Coverslips were mounted with Vitrogel (HM-VI-A500, ErgoProduction, Saint Petersburg, Russia). Visualization was conducted by light microscopy (AxioImager.A1 microscope and EC Plan-Neofluar 20×/0.50 or EC Plan-Neofluar 40×/0.75 M27 objectives, Carl Zeiss, Oberkochen, Germany).

## Figures and Tables

**Figure 1 ijms-24-01959-f001:**
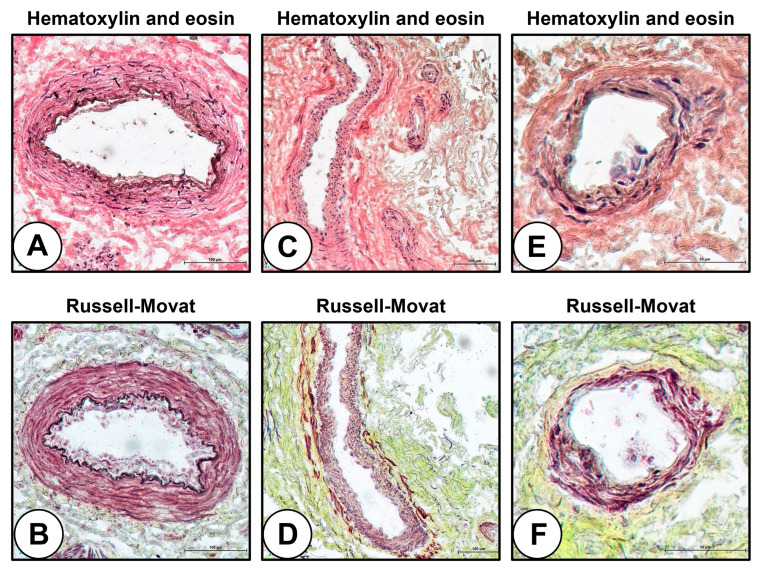
VV types with the hSV. (**A**) Arteriole, H&E staining; (**B**) Arteriole, Russell–Movat’s pentachrome staining; (**C**) Venule, H&E staining; (**D**) Venule, Russell–Movat’s pentachrome staining; (**E**) Capillary, H&E staining; (**F**) Capillary, Russell–Movat’s pentachrome staining. Magnification: ×400.

**Figure 2 ijms-24-01959-f002:**
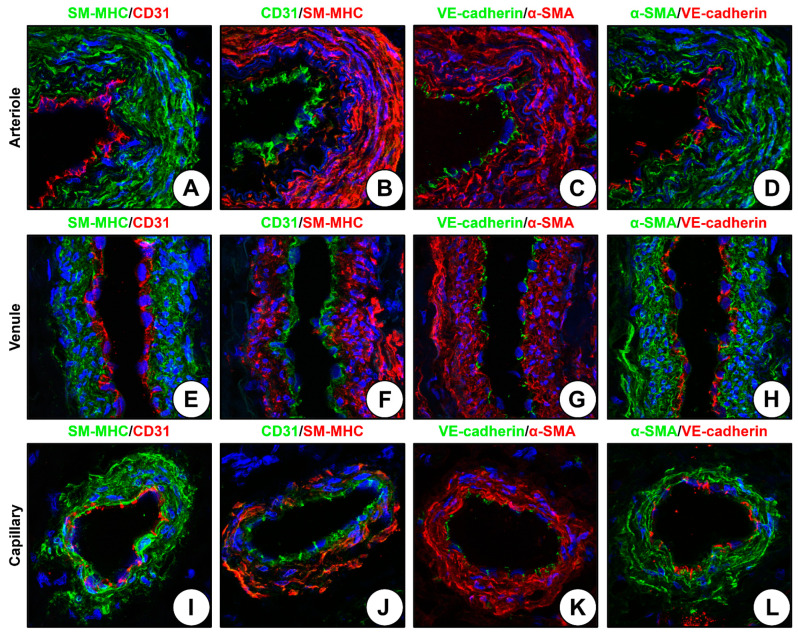
Comparison of EC and VSMC markers in the VV (hSV) staining setting. (**A**) Arteriole, SM-MHC (rabbit) and CD31 (mouse) staining; (**B**) Arteriole, CD31 (rabbit) and SM-MHC (mouse) staining; (**C**) Arteriole, VE-cadherin (rabbit) and α-SMA (mouse) staining; (**D**) Arteriole, α-SMA (rabbit) and VE-cadherin (mouse) staining; (**E**) Venule, SM-MHC (rabbit) and CD31 (mouse) staining; (**F**) Venule, CD31 (rabbit) and SM-MHC (mouse) staining; (**G**) Venule, VE-cadherin (rabbit) and α-SMA (mouse) staining; (**H**) Venule, α-SMA (rabbit) and VE-cadherin (mouse) staining; (**I**) Capillary, SM-MHC (rabbit) and CD31 (mouse) staining; (**J**) CD31 (rabbit) and SM-MHC (mouse) staining; (**K**) VE-cadherin (rabbit) and α-SMA (mouse) staining; (**L**) α-SMA (rabbit) and VE-cadherin (mouse) staining. Magnification: ×400.

**Figure 3 ijms-24-01959-f003:**
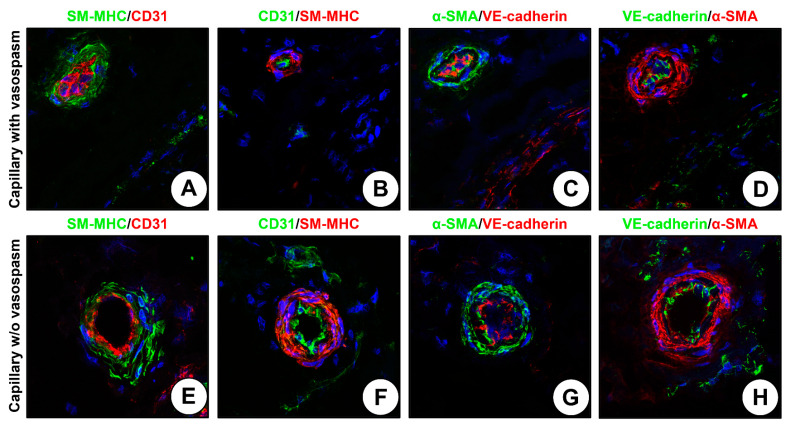
Staining of capillaries with and without vasospasm. (**A**) Capillary with vasospasm, SM-MHC (rabbit) and CD31 (mouse) staining; (**B**) Capillary with vasospasm, CD31 (rabbit) and SM-MHC (mouse) staining; (**C**) Capillary with vasospasm, α-SMA (rabbit) and VE-cadherin (mouse) staining; (**D**) Capillary with vasospasm, VE-cadherin (rabbit) and α-SMA (mouse) staining; (**E**) Capillary without vasospasm, SM-MHC (rabbit) and CD31 (mouse) staining; (**F**) Capillary without vasospasm, CD31 (rabbit) and SM-MHC (mouse) staining; (**G**) Capillary without vasospasm, α-SMA (rabbit) and VE-cadherin (mouse) staining; (**H**) Capillary without vasospasm, VE-cadherin (rabbit) and α-SMA (mouse) staining. Magnification: ×400.

**Figure 4 ijms-24-01959-f004:**
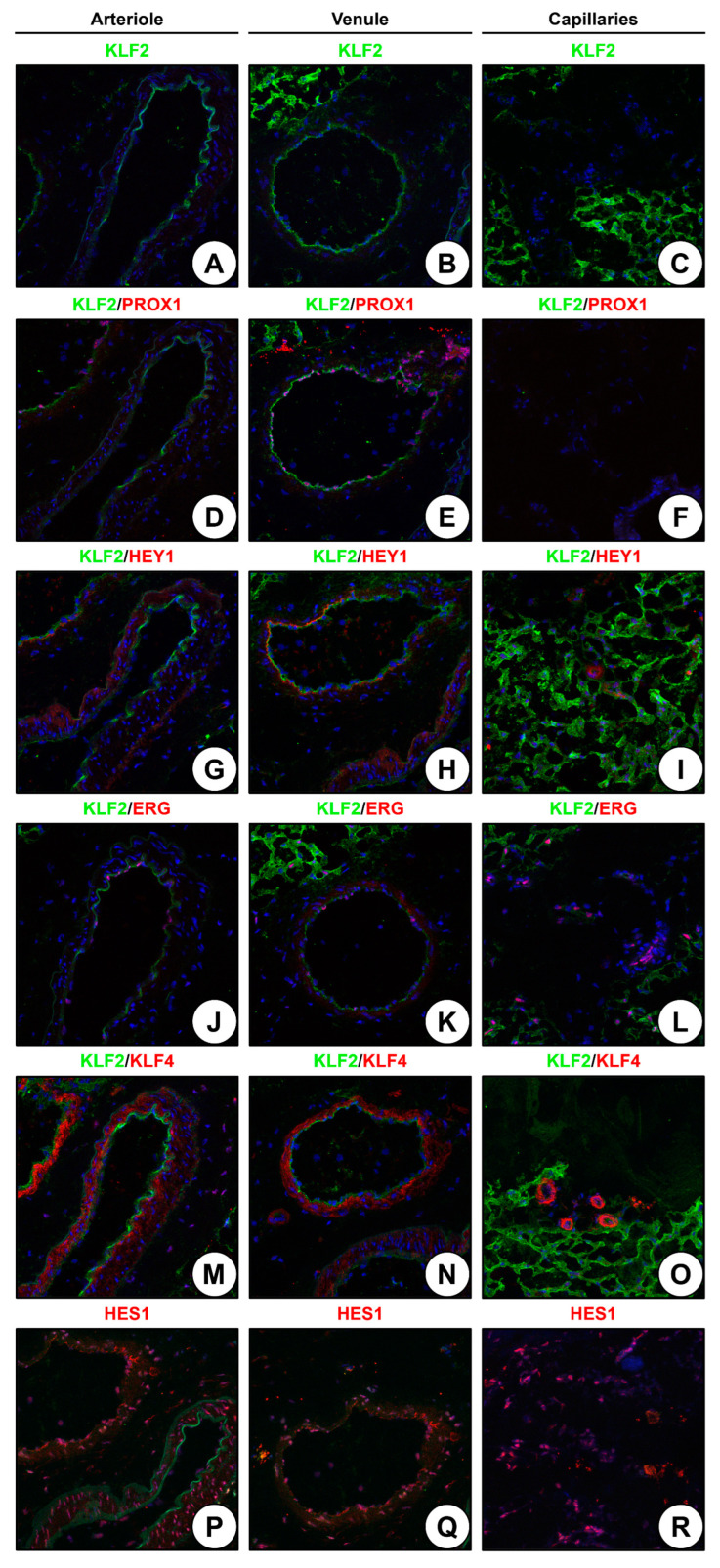
Immunofluorescence staining of rAA arterioles, venules, and capillaries. (**A**) Arteriole, KLF2 staining; (**B**) Venule, KLF2 staining; (**C**) Capillary, KLF2 staining; (**D**) Arteriole, KLF2 and PROX1 staining; (**E**) Venule, KLF2 and PROX1 staining; (**F**) Capillary, KLF2 and PROX1 staining; (**G**) Arteriole, KLF2 and HEY1 staining; (**H**) Venule, KLF2 and HEY1 staining; (**I**) Capillary, KLF2 and HEY1 staining; (**J**) Arteriole, KLF2 and ERG staining; (**K**) Venule, KLF2 and ERG staining; (**L**) Capillary, KLF2 and ERG staining; (**M**) Arteriole, KLF2 and KLF4 staining; (**N**) Venule, KLF2 and KLF4 staining; (**O**) Capillary, KLF2 and KLF4 staining; (**P**) Arteriole, HES1 staining; (**Q**) Venule, HES1 staining; (**R**) Capillary, HES1 staining. Magnification: ×400.

**Figure 5 ijms-24-01959-f005:**
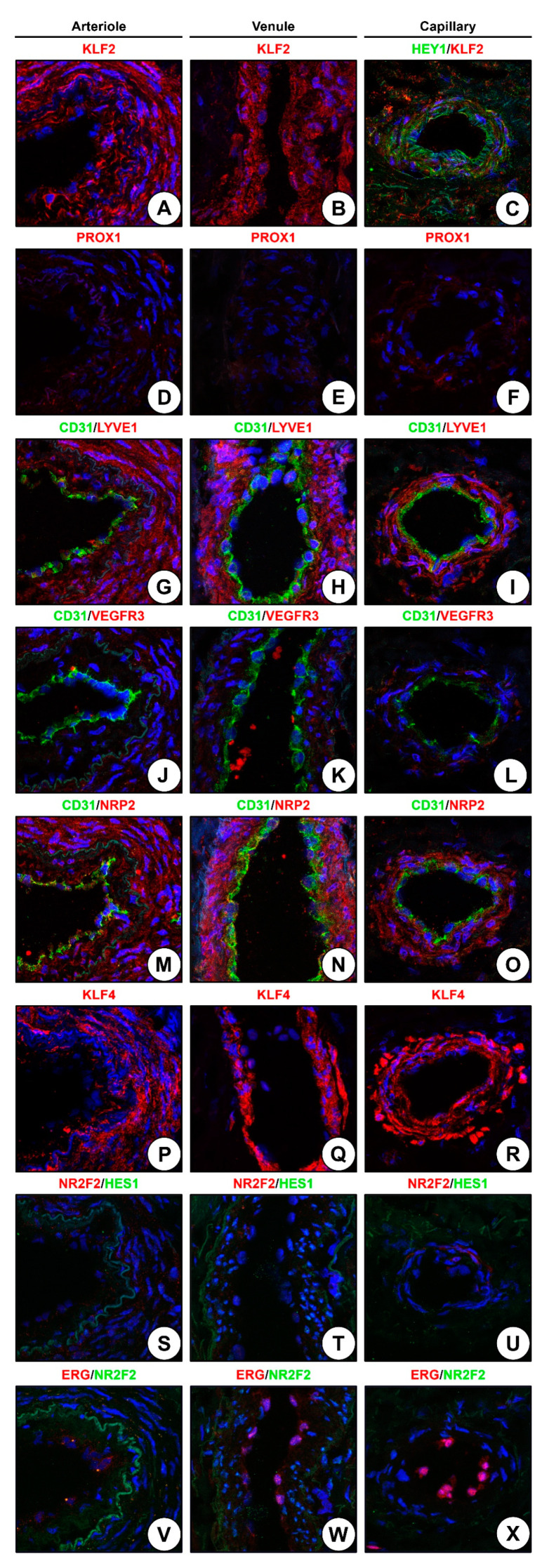
Immunofluorescence staining of hSV arterioles, venules, and capillaries. (**A**) Arteriole, KLF2 staining; (**B**) Venule, KLF2 staining; (**C**) Capillary, HEY1/KLF2 staining; (**D**) Arteriole, PROX1 staining; (**E**) Venule, PROX1 staining; (**F**) Capillary, PROX1 staining; (**G**) Arteriole, CD31 and LYVE1 staining; (**H**) Venule, CD31 and LYVE1 staining; (**I**) Capillary, CD31 and LYVE1 staining; (**J**) Arteriole, CD31 and VEGFR3 staining; (**K**) Venule, CD31 and VEGFR3 staining; (**L**) Capillary, CD31 and VEGFR3 staining; (**M**) Arteriole, CD31 and NRP2 staining; (**N**) Venule, CD31 and NRP2 staining; (**O**) Capillary, CD31 and NRP2 staining; (**P**) Arteriole, KLF4 staining; (**Q**) Venule, KLF4 staining; (**R**) Capillary, KLF4 staining; (**S**) Arteriole, NR2F2 and HES1 staining; (**T**) Venule, NR2F2 and HES1 staining; (**U**) Capillary, NR2F2 and HES1 staining; (**V**) Arteriole, ERG and NR2F2 staining; (**W**) Venule, ERG and NR2F2 staining; (**X**) Capillary, ERG and NR2F2 staining. Magnification: ×400.

**Figure 6 ijms-24-01959-f006:**
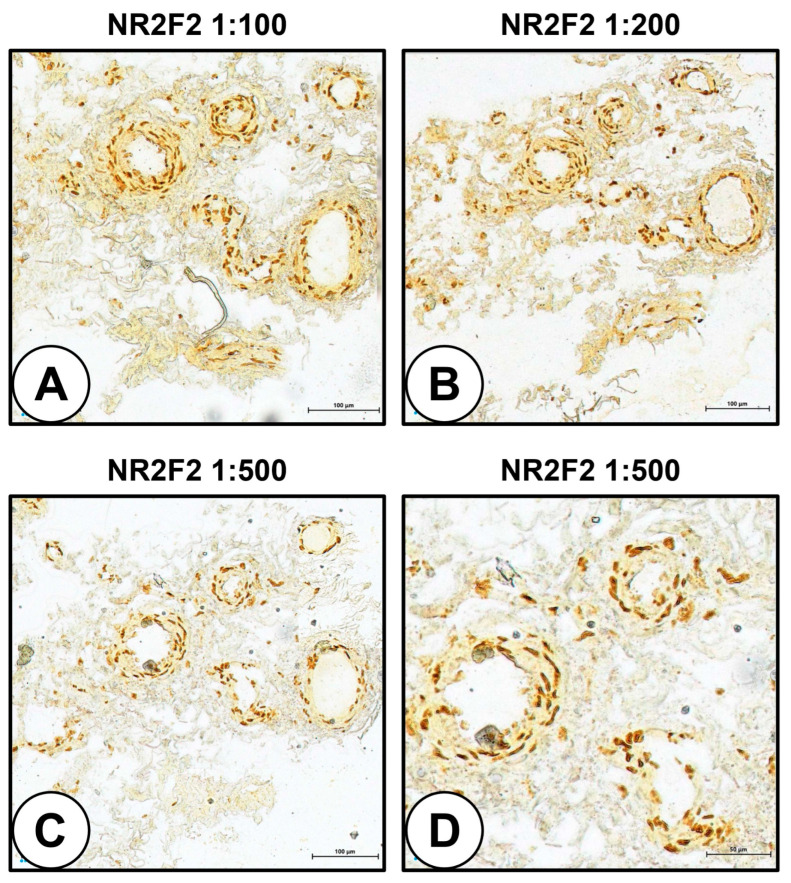
Immunohistochemical staining of hSV for NR2F2. (**A**) NR2F2 staining, antibody dilution 1:100; (**B**) NR2F2 staining, antibody dilution 1:200; (**C**) NR2F2 staining, antibody dilution 1:500, magnification: ×200; (**D**) NR2F2 staining, antibody dilution 1:500, magnification: ×400.

## Data Availability

No new data were created or analyzed in this study. Data sharing is not applicable to this article.

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
