# Peer review of "Endothelial Cell Markers Are Inferior to Vascular Smooth Muscle Cells Markers in Staining Vasa Vasorum and Are Non-Specific for Distinct Endothelial Cell Lineages in Clinical Samples"

_ijms, 2023, doi:10.3390/ijms24031959_

Round 1
Reviewer 1 Report
The results of this study are convincing, however, the samples are frozen sections. Could the authors mention about the results in paraffin sections. Or give a disscussion while the samples are paraffin sections.
On the other hand, pictures of HE staining could be present to demonstrate the pathological orientation and make sure the localization of the specific staining.
Author Response
We sincerely thank the reviewer for the constructive criticism and valuable notes, which collectively helped us to improve the paper. Please see the attachment.

Reviewer 2 Report
The authors performed a combined immunostaining of rat abdominal aorta (rAA) and human saphenous vein (hSV) for various EC or vascular smooth muscle cell (VSMC) markers and found that the latter (e.g., alpha-smooth muscle actin (α-SMA) or smooth muscle myosin heavy chain (SM-MHC)) ensure several-fold higher signal-to-noise ratio irrespective of the primary antibody origin, fluorophore, or Vasa Vasorum type (arterioles, venules, or capillaries).
The study is novel and interesting.
- Introduction is clear.
- Methods are very well detailed
- Results are relevant
- Discussion is full of adequate critical insight.
Overall considered it is an excellent article and the authors should be congratulated.
Author Response
We sincerely thank the reviewer for the high evaluation of our paper. Please see the attachment.
